# Behavioral Responses to Body Position in Bees: The Interaction of *Apis mellifera* and *Lithurgus littoralis* in Prickly Pear Flowers

**DOI:** 10.3390/insects13110980

**Published:** 2022-10-26

**Authors:** Ariadna I. Santa Anna-Aguayo, Edmont Celis-López, Colleen M. Schaffner, Jordan Golubov, Luis E. Eguiarte, Gabriel Arroyo-Cosultchi, Claudia Álvarez-Aquino, Zelene Durán-Barradas, Armando J. Martínez

**Affiliations:** 1Instituto de Neuroetología, Universidad Veracruzana, Dr. Luis Castelazo Ayala S/N Colonia Industrial Animas, Xalapa-Enríquez 91190, Veracruz, Mexico; 2School of Humanities & Social Sciences, Psychology Department, Adams State University, Edgemont Blvd. 208, Alamosa, CO 81101, USA; 3Departamento El Hombre y Su Ambiente, Universidad Autónoma Metropolitana, Calzada Del Hueso 1100, Colonia Villa Quietud, Ciudad de Mexico 04960, Coyoacán, Mexico; 4Laboratorio de Evolución Molecular y Experimental, Departamento de Ecología Evolutiva, Instituto de Ecología, Universidad Nacional Autonoma de México, Circuito Exterior Junto al Jardín otánico Exterior, C.U., Apartado Postal 70-275, Ciudad de México 04510, Coyoacán, Mexico; 5Departamento de Ecología y Recursos Naturales, Facultad de Ciencias, Universidad Nacional Autónoma de México, Ciudad de México 04510, Mexico; 6Instituto de Investigaciones Forestales, Universidad Veracruzana, Parque Ecológico “El Haya” Colonia Benito Juárez, Apdo. 551, Xalapa 91001, Veracruz, Mexico; 7Red de Manejo Biotecnológico de Recursos, Instituto de Ecología, A.C., Carretera Antigua a Coatepec 351, Xalapa 91070, Veracruz, Mexico

**Keywords:** aggression, bees, behavioral response, cactus, flowers, visual information

## Abstract

**Simple Summary:**

Visual information transmitted by body position modulates the interaction and behavior of con- and hetero-specific bees during floral visits of the prickly pear *Opuntia huajuapensis*. Dummy model bees in the feeding or horizontal positions on flowers do not hinder *Apis mellifera* visitation. This exotic species reacted faster to the dummy models of native and conspecifics when in alert or horizontal positions. In contrast, native male *Lithurgus littoralis* spent more time displaying aggressive behaviors towards the dummy model in the alert or horizontal positions but showed a positive response of native female bees towards the dummy of the exotic *A. mellifera* when found in the feeding position. Experimental body position during floral visitation provides a visual cue that modifies bee behavior, which in turn determines access to floral resources.

**Abstract:**

The behavior of bees is modulated by the presence of other bees and potentially by the visual information transmitted by the different body positions of bees while visiting flowers. We tested whether bee body position promoted the attraction and/or antagonistic behavior of con- and hetero-specific bees that interacted on prickly pear flowers of *Opuntia huajuapensis*. To test this, we placed dummy model bees of *Apis mellifera* and the native *Lithurgus littoralis* in flowers in three common body positions: alert, feeding, and horizontal. The results showed that dummy model bees in feeding and horizontal body positions attracted *A. mellifera* bees, while the alert position attracted native male *L. littoralis*. Male *L. littoralis* bees spent more time attacking model bees in horizontal and alert positions. The position of dummy bees also influenced response times. Bees of *A. mellifera* responded fastest to *L. littoralis* in the alert and horizontal position, male *L. littoralis* responded fastest to con-specific bees in the alert and feeding position, and female *L. littoralis* responded fastest to *A. mellifera* bees in the feeding position. *A. mellifera* reacted fastest to their con-specific bees in the alert and horizontal body positions. Our results demonstrate, for the first time in bees, that the position of individuals on a floral resource provides important visual information that modulates bee behavior, and illuminates aspects that likely have implications for bees in access to floral resources.

## 1. Introduction

Visual cues affect the decisions made by bees in terms of the time and direction of behavior (e.g., searching, foraging, and defense). The presence of con- or hetero-specific bees in flowers transmits visual information on the availability of floral resources to other bees among other signals [1,2,3,4,5], so access to floral resources can promote interactions and aggressive behavior among bees [6,7]. Some species assume particular defensive or alert body positions [8] that differ from when feeding, making them vulnerable to con- or hetero-specific aggression that may reduce or increase antagonistic confrontations between individuals [9].

In the case of bees, they display certain body positions to defend nests [10,11,12] and floral resources [13,14]. This arises because bees can recognize the pattern and orientation of objects and can therefore modulate their behavior according to a visual cue [15,16,17]. Body position displays of bees while visiting flowers thus provide information that can potentially affect the attraction or aggressive behavioral responses of other con- and hetero-specifics.

However, to date, the relationship between bee body position on floral resources and the behavioral responses of other bees has not been explored. In particular, there are no experimental studies on how native bees respond to introduced bee species according to the body position displayed by con- and hetero-specific when using the same floral resource.

Recently, Tenorio-Escandón et al. [18] reported that for ca. 80% of the species of the genus *Opuntia*, we lack knowledge of their floral visitors and know even less on plant-visitor interactions and visitor–visitor behavior. Santa Anna-Aguayo et al. [19], in flowers of *Opuntia huajuapensis* Bravo (Cactaceae), showed that 7% of bees were observed in the “collecting nectar position”, 11% in a “horizontal position”, and 9% showed a “warning body position”, while the remaining 73% exhibited flying or aggressive behaviors. Bees are most vulnerable to aggression when they are feeding, whereas horizontal positions indicate rest and alert postures that ultimately represent a defensive condition [20].

We experimentally imitated the three commonly found body positions (alert, feeding, and horizontal) used by native *Lithurgus littoralis* (Cockerell) (Hymenoptera: Megachilidae) and introduced *Apis mellifera* (L.) (Hymenoptera: Apidae) bees when visiting prickly pear flowers containing model bees of each species. The aim of this study was to assess whether different body positions that are encountered by con- and hetero-specific bees on flowers modify visiting behavior and/or aggressive interactions of the two bee species. We proposed the hypotheses that bees exhibiting a vulnerable body position (such as feeding) would be prone to aggression, and that there are differences between *A. mellifera*, as well as the female and male of native bees *L. littoralis*. In addition, the latency (duration of between the stimulus and the first behavioral response) of detection in a floral resource by *A. mellifera* and both sexes of native bees *L. littoralis* would be different between bee model positions.

## 2. Materials and Methods

Experiments were carried out at a site locally known as “malpaís” (vegetation growing on a volcanic lava flow) that is part of the Tehuacan desert in the states of Veracruz and Puebla in Mexico [21] (l9°36′28.42″ N, 97°22′55.74″ W, 2300 asl Figure 1) and experiments were set up in 2018. The site has a semi-arid climate with an annual temperature range of 12 °C to 18 °C and an average rainfall of 500 to 600 mm (Service Meteorologic National, 2021, https://smn.conagua.gob.mx (accessed on 1 September 2021). The vegetation is dominated by *Nolina parviflora* (Kunth) Hemsl., *Yucca periculosa* Baker, *Opuntia huajuapensis* Bravo, *Agave obscura* Schiede ex Schltdl., *A*. *applanta* Lem. ex Jacobi, *Mammillaria discolor* Haw., *M. magnimmama* Haw., *Bouvardia scabrida* (Cav.) Schltdl., *B*. *longiflora* (Cav.) Kunth, *B*. *ternifolia* (Cav.) Schltdl., and *Dalea bicolor* Humb. and Bonpl. ex Willd. [22,23]. The first seven species are the most abundant and conspicuous in the plant community.

### 2.1. Study System

The prickly pear cactus (*Opuntia huajuapensis*) is abundant in the study area. It blooms from May to June and flowers remained open for 1.4 days on average [19]. The flowers are yellow with a diameter of 5-6 cm and are located from 1 to 6 on the margin of the cladodes [24]. Each plant can also have up to six flowers in anthesis per day which are visited by several insect species [25]. Of all floral visitors, the honeybee *A. mellifera* and the native bee *L. littoralis* are the most common [19]. Hereafter, males of *L. littoralis* will be referred as native males, female *L. littoralis* will be referred as native females, and females of *A. mellifera* forage will be referred to as honey bees.

Within the area, there were feral colonies of *A. mellifera* and females forage on different species over the year, but they preferentially feed from the pollen and nectar of *O. huajuapensis* during May and June. *A. mellifera* (1.2–2.0 cm long) presents body hairs not directly used to collect pollen that is collected in corbiculae. During the blooming period, both sexes of *L. littoralis* also feed on *O. huajuapensis* [19] that can be easily distinguished from the honey bee because they become black with white scopa in the ventral part of the abdomen (modified hairs used in pollen transport). *Lithurgus littoralis* are sexually dimorphic, as females are larger (1.3–1.6 cm) than males (1.0–1.3 cm), and they have horn-like structures on the head and collect pollen that they keep in the ventral scopa, unlike males.

### 2.2. Experimental Design

Dummy model bees were manufactured using plaster molds with an odorless dough and water-based paint to mimic the color, morphology, and average size of each bee species [26]. Moreover, considering the size from a sample of 50 bees of each species (100 bees in total), the dummy models were created with an average size of 1.34 cm ± 0.01 SE. These dummy models were then mounted on entomological pins to place inside flowers (Appendix A).

The behavioral effect of con- and heterospecific bees to the three different body positions (alert, feeding, or horizontal) on prickly pear flowers was evaluated for 10 consecutive days at the end of May 2018 during the peak of the flowering period. The dummy model, either *A. mellifera* or native (here after referred to as the type of dummy model), served as one factor in the experiment. In addition, we applied 3 position conditions in independent flowers: (1) upright-positioned dummy models (head up apparently alert or guarding resource), (2) vertically placed dummy models (head facing down imitating feeding behavior (hereafter referred to as the feeding body position), and (3) horizontally placed dummy models (simulating a resting position) (Figure 2). Each flower was used only once, such that encounters to a specific flower assumed a naïve understanding of the visitor regarding the presence of individuals in the flower.

The characteristics of the flowers used in the treatments were standardized (full anthesis with extended petals, receptive stigma, dehiscent anthers, no florivory, and similar size and height). For each flower, size was estimated with the equation of the ellipse (*M* ± *SE* = 16 ± 5 cm^2^) and the height of the flowers above the ground was *M* ± *SE* = 56 ± 10 cm (*n* = 180 flowers). We also selected plants that had a cladode with a single flower and tagged as many individuals as possible to obtain flowers under the conditions indicated above.

A total of 180 observations were made on independent flowers and plants (each plant and flower were used only once). Treatments were applied at random between 1000 and 1100 h to minimize the effect of foraging time. Each flower with its treatment was filmed for 3 min with a Sony Handycam DCR-DVD610 using a 10× zoom with the camera located 1 m away from the flower to avoid interference with visitor behavior [19,26,27].

### 2.3. Environmental Variables

To ensure that the impact of micro-environmental variables (temperature, humidity, and wind velocity) was negligible on the experiment, we recorded the environmental variables at the flower for the duration of the video (two recordings in each of the 180 flowers) with a portable weather tracker (Kestrel 4000; Nielsen-Kellerman, McKellar, ACT, Australia). The average temperature was *M* ± *SE* = 25 ± 1.0 °C, the humidity was *M* ± *SE* = 33 ± 0.4 %, and the wind velocity was *M* ± *SE* = 1.3 ± 0.08 m/s). Thus, video filming sessions occurred in favorable weather conditions for insect activity.

### 2.4. Variables

The variables in each filmed video were registered for each of the two species: (1) the number and sex of visitors of the native bee; (2) the duration of floral visit (contact with some structure of the flower); (3) the frequency of aggressive behaviors (which included fighting, biting, body contact, lunging, or approaching another bee in or close to the filmed flower); (4) the duration of aggression; and (5) the latency of response (duration in seconds from the placement of the treatment dummy to the first behavioral record). 

We analyzed videotapes using image-editing software (Windows Media Player, InterVideo WinDVD). The number of bees per treatment was independent between treatments because only the first behavioral event was registered for each flower [27,28].

### 2.5. Statistical Analysis

All response variables, except for latency, were transformed to proportions to ease variable interpretation and satisfy the assumptions of normality and homogeneity of variances [29]. The variables were analyzed with generalized linear models (GLMs) with a two-factor design where one factor was the type of dummy model (*A. mellifera* or native bee) and the second factor was the body position (alert, feeding, and horizontal) and the respective interaction between the dummy model and body position factors. The latency variable was analyzed with the same GLM model structure, but we used a Poisson distribution (which assumes that the variance is equal to the mean) to adjust error type with a logarithmic link function [30,31] and multiple contrasts tested all possible pairwise comparisons. All analyses were carried out with R library “stats” (R Core Team 2017).

## 3. Results

### 3.1. Number of Visitors

A total of 553 bees were recorded across the different experimental conditions. The number of individuals varied between species: more native bees (*L. littoralis*) visited flowers (355 individuals, *M* ± *SE* = 1.97 ± 2.29 per flower) than *A. mellifera* (194 individuals, *M* ± SE = 1.07 ± 1.22 per flower). 

The percentage of *A. mellifera* individuals attracted to flowers differed depending on the type of dummy model (*F*_2,174_ = 29.80; *p* < 0.0001). More *A. mellifera* bees arrived at flowers with conspecific than heterospecific models (Figure 3A). There were also differences depending on the type of body position (*F*_2,174_ = 28.64; *p* < 0.0001), as more *A. mellifera* bees arrived at flowers with models in horizontal and feeding positions than on an alert position (Figure 3B), and the interaction between the type of dummy model × body position was not significant (*F*_2,174_ = 2.30; *p* = 0.10) (Appendix A), which means that the resulting behavioral responses were consistent across the different types of bee interaction.

The number of *L. littoralis* events was also different between the types of dummy model (*F*_1,174_ = 28.13; *p* < 0.0001) since more bees arrived to flowers with conspecific models (72 ± 3 *SE*) than heterospecific models (*M* ± *SE* = 42 ± 3). Similarly, there were differences depending on the type of body position (*F*_2,174_= 3.31; *p* = 0.03), but in contrast to *A. mellifera*, more *L. littoralis* arrived at flowers with dummy models in an alert position (*M* ± *SE* = 66 ± 5) than in the feeding position (*M* ± *SE* = 57 ± 5), and in less proportion to horizontal models (*M* ± *SE* = 48 ± 5). The dummy model × bee body position interaction was non-significant (*F*_2,174_ = 3.31; *p* = 0.04), also suggesting consistent behavioral responses to bee presence and position.

### 3.2. Number of Native Bees Per Sex

The sexual proportion of *L. littoralis* was biased towards males (303 male individuals (92.2%) (*M* ± *SE* = 1.68 ± 2.03 per flower) and 22 female individuals (6.7%) (*M* ± *SE* = 0.1 ± 0.37 per flower)). A total of 30 events of *L. littoralis* were omitted from the analysis because sex could not be assigned with certainty.

The number of trials scored with native males differed depending on the dummy model species on the flowers (*F*_1,174_= 28.30; *p* < 0.0001), and a higher percentage of males was observed on treatments with conspecific models than heterospecific models (Figure 3C). *Post-hoc* comparisons showed that there were also more visits to dummy models in the alert than in resting or feeding positions (*F*_2,174_ = 3.60; *p* = 0.04, Figure 3D). The model type × position interaction was not significant (*F*_2,174_ = 2.10; *p* = 0.09), meaning that the number of trials scored with native males is a parallel response between the effect of the dummy model species and the model’s three body positions.

Females of *L. littoralis* did not respond to the type of dummy model (*M* ± *SE* = 4 ± 1, *F*_1,174_ = 0.64; *p* = 0.40) or to the three body positions of bees (*M* ± *SE* = 4 ± 2, *F*_2,174_ = 2.30; *p* = 0.10). In addition, the behavioral response was independent of a crossover effect of both factors, as shown by the non-contribution of the bee type × bee body position interaction (*F*_2,174_ = 0.10; *p* = 0.80).

### 3.3. Number of Bees Visiting Flowers

From the total of bees with visiting behavior to prickly pear flowers (148 visit events), 83% (123 events) were made by individuals of *A. mellifera*, and 9% (13 events) were made by males of *L. littoralis* and 8 % (12 events) were made by females of *L. littoralis*.

When comparing the average percentages of visits with dummy models, *A. mellifera* made no distinction between the *A. mellifera* model and the native bee model (Table 1a). Furthermore, the number of *A. mellifera* visitors was also not significant between the three body position treatments (Table 1a) or by the type of bee × body position of dummy bee (*F*_2,174_ = 1.70; *p* = 0.30). This suggests that the behavior *A. mellifera* is directly related to the dummy model of each species and the body position, regardless of the effect of the interaction. 

Visiting events of native male bees were more common when dummy conspecific bees were present on the flowers than heterospecific dummies (Table 1b). However, native males did not show differences in visiting patterns across the three body positions (Table 1b) or in the interaction (*F*_2,174_ = 0.9; *p* = 0.3). This shows than the native bees responded to the presence of conspecific bees on the flowers, regardless of the bee body position. 

Likewise, female native bees did not differ in their pattern of visiting flowers when they had dummy model species, as shown in the *A. mellifera* model and the *L. littoralis* model (Table 1c), or between bee body positions (Table 1c), and there was no effect of the interaction of both factors (*F*_2,174_ = 0.20; *p* = 0.70). This indicates that the visits of female native bees are not affected by the presence of the dummy model bees. 

### 3.4. Duration of Floral Visits

The average percentage of time that the *A. mellifera* bees visited flowers did not differ between the types of dummy model (*p* = 0.20, Table 2a) or as a function of the dummy model body position (Table 2a). Furthermore, there was no effect of the interaction (*F*_2,174_ = 1.50; *p* = 0.30). This means that the time that *A. mellifera* bees remained did not vary due to the effect of dummy model species and bee body position.

Males of *L. littoralis* spent more time on flowers with conspecific models than with *A. mellifera* models (Table 2b), but floral visiting time by males did not differ across body position (Table 2b) or as a function of the interaction of the two factors (*F*_2,174_ = 0.6; *p* = 0.50). Native females did not differ in the duration of visits to flowers between the dummy model types of *A. mellifera* and *L. littoralis* (Table 2c), and the time spent was not affected by body position (Table 2c) or by the interaction (*F*_2,174_ = 0.10; *p* = 0.80). Therefore, the average percentages of time of native male and female bees visiting flowers was independent of the presence of dummy model species and bee body position.

### 3.5. Aggressive Behavior

In all model treatments, only native bees displayed aggressive behavior. From a total of 217 native bees that were aggressive to models, 212 were males and 5 were females. The mean number of events with males displaying aggressive behavior differed between the types of dummy models (*F*_1,174_= 13.60; *p* < 0.0001), as *L. littoralis* males were more aggressive to conspecifics than to the *A. mellifera* dummy models (Figure 4A). In addition, the percentage of events with males that displayed aggressive behavior to the dummy models differed depending on the position (*F*_2,174_= 8.70; *p* < 0.0001). Models in the horizontal and alert positions received more aggressive responses (Figure 4B) than the feeding position. However, the interaction between the type of dummy model and body position was not significant (*F*_2,174_= 0.10; *p* = 0.80). This shows that the number of aggressions carried out by bees on dummy model species and bee body position was independent and specific to each of these treatment factors.

The percentage of time spent by males displaying aggressive behavior differed across the types of dummy models (*F*_1,174_ = 12.20; *p* < 0.0001), as *L. littoralis* males spent more time being aggressive with conspecific than with heterospecific models (Figure 4C). In addition, the position of models influenced the duration of aggressive behavior (*F*_2,174_ = 8.40; *p* < 0.0001). Models in the alert and horizontal position had longer aggressive incidents than models in a feeding position (Figure 4D). There was no interaction effect between the two factors (*F*_1,174_ = 0.05; *p* = 0.90). Likewise, the time of the aggressions of the bees towards the dummy model species and the bee body position also delimited independent effects without implying a crossing of both factors.

### 3.6. Latency

*Apis mellifera* detected the presence of conspecifics faster than heterospecific dummy models (χ^2^ = 170.10, *p* < 0.0001). Latency also differed when comparing body position (χ^2^ = 11.40; *p* < 0.0001), as the models in horizontal positions were detected faster than when they were in alert or feeding positions. The interaction between the type of model × position was also significant (χ^2^ = 168.20; *p* < 0.0001), as *A. mellifera* bees did not differ in their latency to respond to models in the alert position, regardless of species, but had a slower latency to respond to conspecific compared with heterospecific models when they were in the horizontal and feeding position (Figure 5A). This indicates that *A. mellifera* bees have a slow behavioral response to conspecific dummy models that do not appear to represent a threat.

Male native bees also had different response times to detect dummy bees in flowers (χ^2^ = 171.50; *p* < 0.0001); they reacted faster to conspecifics than to heterospecifics. The position of dummy bees also influenced the response time (χ^2^ = 100.20; *p* < 0.0001), whereby the alert body position led to the fastest response time by males (*M* ± *SE* = 40 ± 0.02 s), followed by horizontal (*M* ± *SE* = 50 ± 0.02 s) and finally feeding position. The interaction between the type of model × body position also contributed to differences in the response time of *L. littoralis* males (χ^2^ = 149.50; *p* < 0.0001), as they responded fastest to feeding and alert conspecific dummy models compared to heterospecific bee models and did not differ in their latency to respond to either model in the horizontal position (Figure 5B).

The latency of female *L. littoralis* bees was faster for conspecific than heterospecific models (χ^2^ = 11.30; *p* < 0.0001). Female bees also responded differently to body position (χ^2^ = 11.10; *p* < 0.0001), in which the horizontal position was perceived faster, than the alert and the feeding positions. The interaction of model type × body position also influenced the latency to respond (χ^2^ = 108.50; *p* < 0.0001), as females responded slowest to conspecific models when they were in the feeding position, but faster to conspecific models when they were in the alert or horizontal position (Figure 5C).

## 4. Discussion

According to our results in the study, species position is an important modulator of the behavioral response between con- and heterospecific bees in flowers. *Apis mellifera* and males of *L. littoralis* were visually attracted to conspecific model bees. Feeding and horizontal dummy models attracted *A. mellifera*, while alert position attracted male native bees. We also found that male *L. littoralis* spent more time attacking dummy model bees in horizontal and alert positions, a possible territorial behavior linked to access to floral resources. Furthermore, the position of dummy models on the flowers of *O. huajuapensis* impacted the latency time of *A. mellifera* that responded very fast to the dummy model of *L. littoralis* when it was in alert and horizontal position.

Bee body position in flowers was a visual factor that modulated con-and heterospecific interactions, a behavior which had not been previously identified, even though studies have in part addressed the issue [1,2,32]. For instance, in birds and crustaceans, the position that individuals adopt when feeding can make them vulnerable to aggression [33,34,35]. 

The presence of other species bees may be a stimulus that acts as a proxy of available flower resources, as *A. mellifera* visited flowers more frequently with conspecific dummy models than heterospecific dummy models. These responses may be learned through the association of the bee’s coloration patterns, which has been observed in other Hymenoptera species that can associate flowers and object color saturation with the availability of food rewards [36,37]. Research has demonstrated that the honeybee and the alfalfa leaf cutting bee (*Megachile rotundata*) can discriminate the visual stimuli of varied shapes and colors and associate them through learning [38]. This capacity for learning is thought to act while foraging, which should allow bees to reduce search time by recognizing flowers that offer a larger reward of nectar or pollen [17,38,39].

In some studies, the co- and heterospecific bee presence was reported as a visual factor associated with food availability [1,2]. In our case, the larger number of *A. mellifera* individuals visiting the flowers with dummy models in feeding and horizontal position might be due to the capacity of *A. mellifera* to associate body posture with potential food availability. Similarly, our results on time in second (latency) indicated that *L. littoralis* female bees spent more time in the flowers when the conspecific dummy in feeding position and we observed them consuming nectar, a behavior that would reduce the energy cost involved in searching and collecting floral resources [40].

The alert position of dummy bee models promoted male *L. littoralis* aggression, which could be associated with a territorial and attack behaviors that activates the innate response of males to aggressively chase conspecifics off flowers for access to females or resources [41]. This information can help reduce the energy and time costs of searching for females, but would increase the risk of intra-sexual competition, a possibility supported by the short duration of aggression to the other body position bee models.

*Lithurgus littoralis* males invested more time engaged in aggressive behavior with dummy conspecifics on flowers of *O. huajuapensis* than *L. littoralis* females or honey bees. The duration of aggressive interactions probably reduced the time males spent collecting pollen and nectar. A similar response was found in the solitary bee *Centris pallida* [41]. There were few aggressive events between *A. mellifera* and the *L. littoralis* dummy, which suggests honeybees tend to avoid aggressive encounters and competition for floral resources. 

Yokoi and Fujisaki [32] recorded that *A. mellifera* avoided visiting flowers of *Rubus hirsutus* when aggressive heterospecific bees were present, which coincides with our earlier findings that *A. mellifera* and female *L. littoralis* avoid interacting with males of *L. littoralis* in flowers of *O. huajuapensis* [19]. 

For *L. littoralis* male bees, the latency to approach models was lower with conspecifics, probably associated with dark object perception [41] or an intra-sexual competition also shown in *C. pallida* [42], *Anthophora plumipes* [43], and *Ptilothrix fructifera* on flowers of *Opuntia* [44]. Furthermore, the fast attraction of *A. mellifera* and both sexes of native bees to dummy individuals of *L. littoralis* was probably due to the color contrast of the body above flowers, which stimulates the visual receptors from a greater distance, as suggested by Lunau [45] and Biesmeijer [46]. 

## 5. Conclusions

*A. mellifera* and native bee *L. littoralis* visually recognized and associated color and body position patterns of con- and heterospecifics in flowers of *O. huajuapensis*. The position of bees on the flowers is visually related to different contexts learned by bees in their interaction with flowers, so visual information they receive from bees present in floral resources would enable con- and heterospecifics to modulate their behavior and optimize access to resources.

## Figures and Tables

**Figure 1 insects-13-00980-f001:**
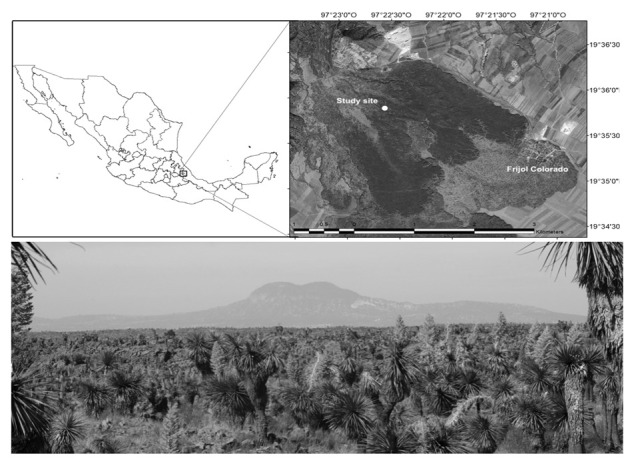
Study area in arid zone showing the lava flow environment near the town of Frijol Colorado (l9°36′28.42″ N, 97°22′55.74″ W). A panoramic view of “malpaís” is also shown with the predominant plant species *Nolina parviflora* and *Opuntia huajuapensis*, a habitat where the honeybee *Apis mellifera* and the native bee *Lithurgus littoralis* develop.

**Figure 2 insects-13-00980-f002:**
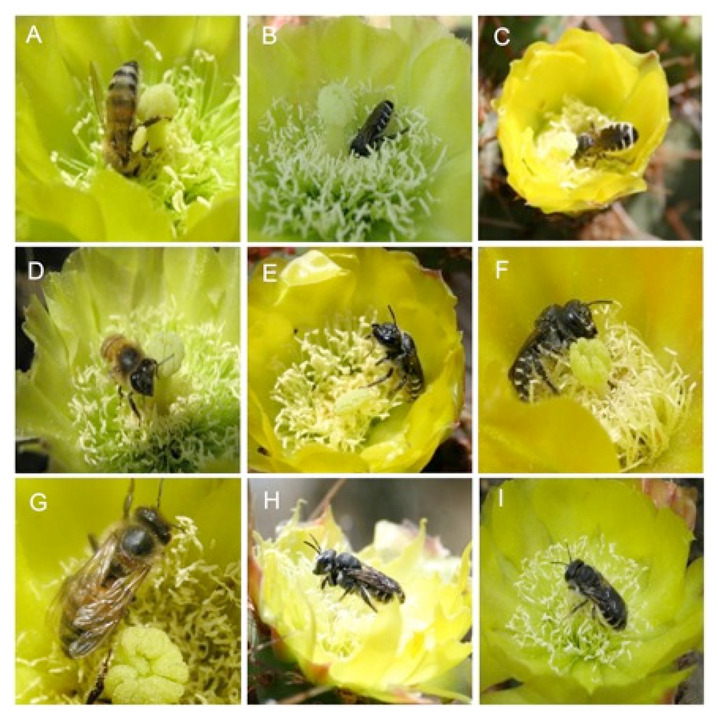
Body position of feeding in *Apis mellifera* bee (**A**), and male (**B**) and female (**C**) of *Lithurgus littoralis.* Alert position in *A. mellifera* (**D**), and male (**E**) and female (**F**) of *L. littoralis.* Horizontal position in *A. mellifera* (**G**), and male (**H**) and female (**I**) of *L. littoralis* bees observed while visiting flowers of *Opuntia huajuapensis*.

**Figure 3 insects-13-00980-f003:**
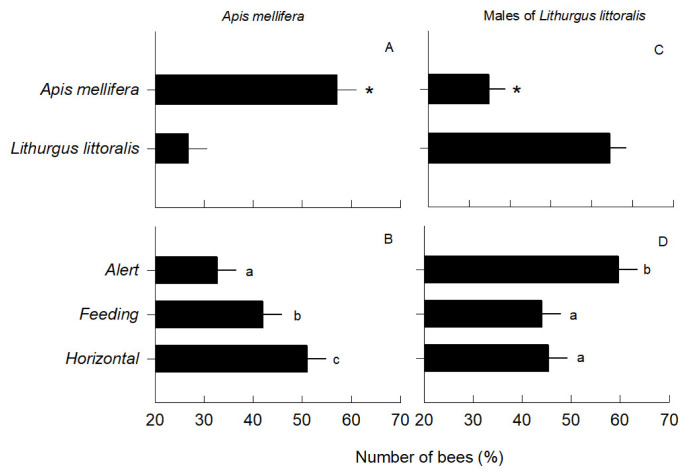
Mean percentage (± *SE*) of the number of events with *Apis mellifera* and *Lithurgus littoralis* males on the flowers of *Opunia huajuapensis* with different model species (**A**,**C**) in three different body positions (**B**,**D**). Each graph shows the effect of the fixed factors of the two-factor GLM, and asterisks and small letters (a, b, and c) on the bars show the significant differences between mean values.

**Figure 4 insects-13-00980-f004:**
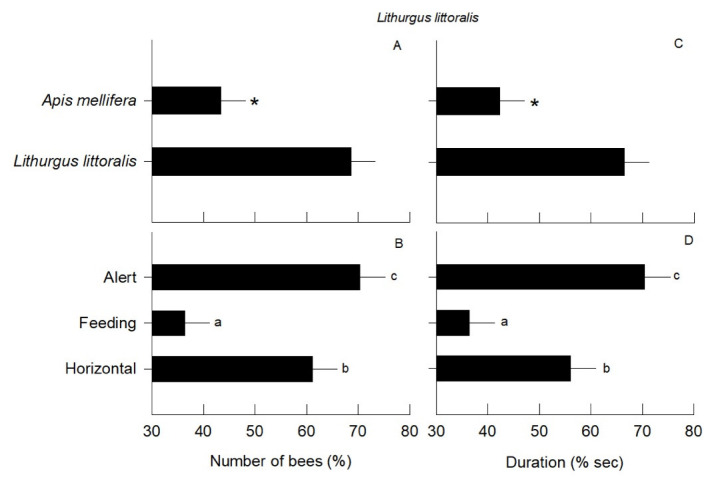
Mean percentage (± *SE*) of the number of male bees of *Lithurgus littoralis* that display aggression in the different dummy model species (**A**) in three different body positions (**B**), as well as the percentage duration of aggression in seconds for the different dummy model species (**C**), and in three different body positions (**D**). Each graph shows the effect of the fixed factors of the two-factor GLM, and asterisks and small letters (a, b, and c) on the bars show the significant differences between mean values.

**Figure 5 insects-13-00980-f005:**
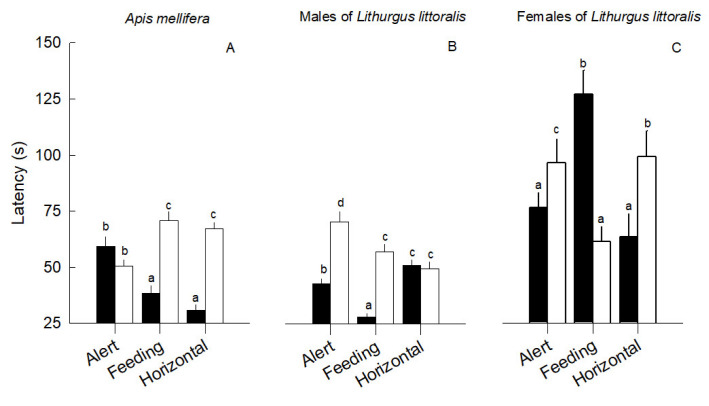
Mean response time in seconds (latency) (*M* ± *SE*) of (**A**) *Apis mellifera*, (**B**) *Lithurgus littoralis* male and (**C**) female to the presence of *A. mellifera* dummy model bees (open bars) and *L. littoralis* dummy model bees (filled bars) for the three body position treatments above flowers of *Opuntia huajuapensis.* Small letters (a, b, c, and d) on the bars show the significant differences between mean values.

**Table 1 insects-13-00980-t001:** Percentages of number of bees visiting flowers (*M* ± *SE*) of *Opuntia huajuapensis* with model bees of *Apis mellifera and the native bee Lithurgus littoralis* (male and female) and comparing between the three body position treatments of the dummy bee.

	Type of Dummy Model		Type of Dummy Models		
	*A. mellifera*	*L. littoralis*	Alert	Feeding	Horizontal
(a) Honey bee	48 ± 5	37 ± 5	35 ± 6	46 ± 6	45 ± 6
	*F*_1,174_ = 2.20, *p* = 0.10		*F*_2,174_ = 1.00, *p* = 0.30		
(b) Native males	8 ± 1	0.6 ± 1	2 ± 2	8 ± 2	2 ± 2
	*F*_1,174_ = 7.60, *p* <0.0001		*F*_2,174_ = 2.20, *p* = 0.1		
(c) Native female	4 ± 2	4 ± 2	6 ± 2	7 ±2	2 ± 2
	*F*_1,174_ = 1.10, *p* = 0.90		*F*_2,174_ = 1.40, *p* = 0.30		

**Table 2 insects-13-00980-t002:** Percentages of time spent on floral visits (*M* ± *SE*) of *Opuntia huajuapensis* with model bees of *A. mellifera* and *L. littoralis* native bees (male and female) and three-body position treatments of dummy bee.

	Type of Dummy Model		Type of Dummy Models		
	*A. mellifera*	*L. littoralis*	Alert	Feeding	Horizontal
(a) Honey bee	47 ± 5	30 ± 5	36 ± 6	45 ± 6	45 ± 6
	*F*_1,174_ = 1.70, *p* = 0.20		*F*_2,174_ = 0.70, *p* = 0.20		
(b) Native male	1 ± 2	7 ± 2	2 ± 2	7 ± 2	3 ± 2
	*F*_1,174_ = 5.30, *p* = 0.02		*F*_2,174_ = 1.10, *p* = 0.10		
(c) Native female	5 ± 2	5 ± 5	6 ± 3	7 ± 3	1 ± 2
	*F*_1,174_ = 0.03, *p* = 0.80		*F*_2,174_ = 1.70, *p* = 0.20		

## Data Availability

The dataset utilized in this study is available upon request.

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
