# Peer review of "Behavioral Responses to Body Position in Bees: The Interaction of *Apis mellifera* and *Lithurgus littoralis* in Prickly Pear Flowers"

_insects, 2022, doi:10.3390/insects13110980_

Round 1

Reviewer 1 Report

I have only minor suggestions to the manuscript :

row 75..why do you indicate Fig. 1 here? you deal with positions of bees ..

row 86 ."..,and different between species and sex." ...the meaning of he sentence is not clear

row 217.."22 individuals (6.7%).." ..it should be specified that it concerns females..?

supplementary videos are very short - what is the meaning to present them?

Author Response

Dear reviewer, we attach the timely response to your comments, adjustments, and corrections, many thanks for your review.

  • Row 75..why do you indicate Fig. 1 here? you deal with positions of bees.

Answer: Done, the indication of Figure 1 was removed, because it did not correspond to the section.

  • Row 86 ...".., and different between species and sex." ...the meaning of he sentence is not clear.

Answer: Done, sentence was adjusted to clear up confusion.

  • Row 217.."22 individuals (6.7%).." ..it should be specified that it concerns females..?

Answer: Done, it was specified that it referred to female bees.

  • Supplementary videos are very short - what is the meaning to present them?

Answer: In order not to locate files with many megabytes, short videos were located. However, longer videos were incorporated to exemplify the visit of Apis mellifera and Lithurgus littoralis in the presence of dummy bees.

Reviewer 2 Report

Peer review of manuscript ID 1979597

Behavioral responses to body position in bees: The interaction of Apis mellifera and Lithurgus littoralis in prickly pear flowers.

Ariadna I. Santa Anna-Aguayo, E. Celis-López, C. M. Schaffner, J. Golubov, L. E. Eguiarte, G. Arroyo-Cosultchi, C. Álvarez-Aquino, Z. Durán-Barradas, and A. J. Martínez

In this manuscript, the authors present their research on the behavioral responses of Apis mellifera, and male and female Lithurgis littoralis individuals to the presence and body position of con- and heterospecific dummy models in flowers of Opuntia huajuapensis. They proposed two hypotheses, 1) bees in a feeding position elicit aggression and that the frequency of these interactions will vary depending on species and sex, and 2) the body position of the dummy models in the prickly pear flowers influences the latency of detection by A. mellifera and L. littoralis individuals. To test their hypotheses, the authors placed con- and heterospecific dummy models into prickly pear flowers in one of three body positions, alert, horizontal, and feeding. The authors recorded the behavioral responses of A. mellifera and L. littoralis individuals to the dummy models.

The authors determined that body position and con- and heterospecific model type are important modulators of behavior in Apis mellifera and Lithurgis littoralis individuals. Apis mellifera and L. littoralis males are more attracted to flowers with conspecific dummy models. Apis mellifera individuals visited flowers with the conspecific dummy model in the feeding and horizontal body positions as these may indicate food availability. Males of L. littoralis displayed more aggression towards the conspecific dummy model in the alert position which could indicate an innate response resulting in eliminating competition. The authors conclude that Apis mellifera and Lithurgus littoralis individuals use visual information to modulate behavior and maximize resource access.

Comments

The experimental design, analysis, and discussion presented in this manuscript are thorough and scholarly. The authors correctly state that little is known about the pollinators of the Opuntia genus. Their research is an important contribution towards understanding the pollination of Opuntia species and interspecific bee interactions at floral resources. My comments regarding the manuscript are focused on minor technical errors.

Line 75 refers to Figure 1 from Santa Anna-Aguayo et al. (2017) paper detailing the body positions of bees visiting flowers of Opuntia huajuapensis. The information cited by the authors is relevant to their research; however, Figure 1 illustrates behavior accumulation curves for A. mellifera and L. littoralis rather than specific behaviors. The authors should correct this minor inconsistency. 

Line 264 should state 3.4. Duration of floral visits.

Line 282 should state 3.5. Aggressive behavior.

Line 310 should state 3.6. Latency

Lines 371 – 373 state that the results suggest that L. littoralis females related feeding position of A. mellifera models with the presence of nectar. This is an interesting suggestion and may have merit, but it is unclear which result forms the basis for this suggestion. Your results do not show a significant behavioral response for L. littoralis females and model type or body position. Please elaborate on this discussion point.

Figure 4 illustrates the mean percentage of the number of male bees of Lithurgus littoralis that display aggression based on con- and hetero-specific model and body position. The caption indicates that the label above (A) should read Males of Lithurgus littoralis rather than Apis mellifera.

Author Response

Dear reviewer, we attach the timely response to your comments, adjustments, and corrections, many thanks for your review.

  • Line 75 refers to Figure 1 from Santa Anna-Aguayo et al. (2017) paper detailing the body positions of bees visiting flowers of Opuntia huajuapensis. The information cited by the authors is relevant to their research; however, Figure 1 illustrates behavior accumulation curves for A. mellifera and L. littoralis rather than specific behaviors. The authors should correct this minor inconsistency.

Answer: Done, the indication of Figure 1 was removed, because it did not correspond to the section.

  • Line 264 should state 3.4. Duration of floral visits.
  • Line 282 should state 3.5. Aggressive behavior.
  • Line 310 should state 3.6. Latency

Answer: Done, fixed the number of each section.

  • Lines 371 – 373 state that the results suggest that L. littoralis females related feeding position of A. mellifera models with the presence of nectar. This is an interesting suggestion and may have merit, but it is unclear which result forms the basis for this suggestion. Your results do not show a significant behavioral response for L. littoralis females and model type or body position. Please elaborate on this discussion point.

Answer: Done, the sentence was corrected referring to the results of the time in second (latency) of the visit of the female bees of L. littoralis, since they spent more time on the flowers when they had the dummy model bees in the feeding position. Also, during that time we observed that the female bees were feeding on nectar. The sentence in sentence in lines 229 and 230 was also corrected, to avoid misinterpretation.

  • Figure 4 illustrates the mean percentage of the number of male bees of Lithurgus littoralis that display aggression based on con- and hetero-specific model and body position. The caption indicates that the label above (A) should read Males of Lithurgus littoralis rather than Apis mellifera.

Answer: Done and captions in Figures 3-5 have been adjusted.
